# Risk Factors for Emergency Department Unscheduled Return Visits

**DOI:** 10.3390/medicina55080457

**Published:** 2019-08-09

**Authors:** Crystal Harn Wei Soh, Ziwei Lin, Darius Shaw Teng Pan, Weng Hoe Ho, Malcolm Mahadevan, Mui Teng Chua, Win Sen Kuan

**Affiliations:** 1Emergency Medicine Department, National University Hospital, National University Health System, Singapore 119074, Singapore; 2Yong Loo Lin School of Medicine, National University of Singapore, Singapore 119228, Singapore; 3Department of Surgery, Yong Loo Lin School of Medicine, National University of Singapore, Singapore 119228, Singapore

**Keywords:** unscheduled return visits, re-attendance, risk management, abdominal pain

## Abstract

*Background and Objectives*: This study aims to identify reasons for unscheduled return visits (URVs), and risk factors for diagnostic errors leading to URVs, with comparisons to data from a similar study conducted in the same institution 9 years ago. *Materials and Methods*: This retrospective study included adult patients who attended the emergency department (ED) of a tertiary hospital in Singapore between January 2014 and June 2014, with re-attendance within 72 h for the same or similar complaint. The primary outcome was wrong or delayed diagnoses. Secondary outcomes include admission to the ED observation unit or ward on return visit. Findings were compared with the previous study performed in 2005 to identify trends. *Results*: Of 67,422 attendances, there were 1298 (1.93%) URVs from 1207 patients (median age 34, interquartile range 24 to 52 years; 59.7% male). The most common presenting complaint was abdominal pain (22.2%). One hundred ninety-one (15.8%) patients received an initial wrong or delayed diagnosis. Factors (adjusted odds ratio; 95% CI) associated with this were: presenting complaints of abdominal pain (2.99; 2.12–4.23), fever (1.60; 1.1–2.33), neurological deficit (4.26; 1.94–9.35), and discharge without follow-up (1.61; 1.1–2.26). Among re-attendances, 459 (38.0%) required admission. Factors (adjusted odds ratio; 95% CI) associated with admission were: male gender (1.88; 1.42 to 2.48); comorbidities of diabetes mellitus (2.07; 1.29–3.31), asthma (5.23; 1.59–17.26), and renal disease (7.48; 2.00–28.05); presenting complaints of abdominal pain (1.83; 1.32–2.55), fever (3.05; 2.10–4.44), and giddiness or vertigo (2.17; 1.26–3.73). There was a reduction in URV rate compared to the previous study in 2005 (1.93% versus 2.19%). Abdominal pain at the index visit remains a significant cause of URVs (22.2% versus 25.1%). *Conclusions*: Presenting complaints of neurological deficits, abdominal pain, fever, and discharge without follow-up were associated with wrong or delayed diagnoses among URVs.

## 1. Introduction

Unscheduled return visits (URVs) to the emergency department (ED) are repeat presentations of patients, discharged from the ED, returning with related presenting complaints. While there is no worldwide consensus on the interval a return visit should fall within [1], studies in North America have commonly used 72 h to define return visits [2]. The proportion of URVs varies (from 0.4% to 43.9%) and is dependent on locality and healthcare system [2]. URVs are a significant problem universally and are monitored as a performance indicator for quality of clinical care in the ED [3,4]. URVs may reflect shortcomings in care provision which may stem from errors in diagnosis, management, and disposition [5]. The proportion of URVs resulting in admission also reflects suboptimal management and may be an indicator of care [6]. Conversely, URVs could be due to reasons independent of quality of care, such as natural progression of disease [7]. URVs have been linked to ED overcrowding [8], which results in protracted waiting time and strain on limited resources [9].

This study aims to determine reasons for ED URVs in a tertiary healthcare institution, identify risk factors contributing to diagnostic errors leading to URVs, and factors leading to admission on repeat visit. Diagnostic error was defined according to the Institute of Medicine (IOM), as “the failure to establish an accurate and timely explanation of the patient’s health problem(s) or communicate that explanation to the patient” [10]. A similar study done in 2005 at the same center [11] allows us to compare trends and evaluate effectiveness of clinical care delivery utilizing better protocols and practices.

## 2. Materials and Methods

This retrospective study involved consecutive patients aged ≥16 years, who presented to the study site ED, a 1225 bed tertiary academic medical center in Singapore, between 1 January 2014 and 30 June 2014. Inclusion criteria for URVs were patients who presented to the ED and re-attended within 72 h of their index visit. The index visit was defined as the initial of two distinct visits made by a patient who returned within 72 h of discharge to the ED. We excluded patients who sought treatment for unrelated complaints during repeat visits, scheduled re-attendances, frequent attenders with >3 ED visits over the preceding 6-month period, and patients who attended due to alcohol intoxication with no other medical complaints. What was considered an “unrelated complaint” was decided a priori. Ambiguous cases were discussed among three independent reviewers (C.H.W.S., Z.L., D.S.T.P.) until consensus was reached.

Medical records were reviewed through ED electronic medical records (ED Web). Variables collected included patient factors (demographics, past medical history, and activities of daily living) and factors regarding index and re-attendance visits (presenting complaint, shift period of presentation, primary doctor designation, presence of senior doctor input, initial and eventual diagnoses, initial and eventual disposition, triage vital signs, hours to re-attendance, reason for re-attendance, eventual length of stay, morbidity, and mortality). In our institution, the majority of patients are seen by junior doctors (residents, medical officers, or locums). A senior doctor in our institution is defined as an emergency medicine senior resident or an attending emergency physician. Patients seen in our institution are triaged based on the Patient Acuity Category (PAC) scale: PAC 1 indicates life-threatening conditions requiring immediate medical attention, PAC 2 signifies serious conditions requiring early medical attention, and PAC 3 indicates acute conditions requiring non-urgent medical attention. Patients triaged to PAC 1 or 2 are reviewed by a senior doctor, whereas patients triaged to PAC 3 are only reviewed by a senior when deemed necessary by the primary physician. Electronic records were supplemented with written records from the medical records office. Ethics approval was obtained from the National Healthcare Group Domain Specific Review Board (DSRB 2014/01209, 18 December 2014) for waiver of informed consent.

Data were collected using a standardized data collection form by three investigators (C.H.W.S., Z.L., D.S.T.P.). To facilitate statistical analyses, presenting complaints were collected individually and then grouped under various clinically-relevant headings. For example, patients who presented with symptoms such as diarrhea and vomiting were grouped under “gastrointestinal symptoms”, symptoms such as cough, rhinorrhea, or sore throat were grouped under “upper respiratory tract infection”, and the main complaint of fever with no other significant symptoms were grouped under “fever” (Appendix A). The primary outcome was wrong or delayed diagnoses, defined as a subsequent diagnosis made at the URV that differs from the initial diagnosis, which was a surrogate measure for diagnostic error. This was based on a review of the patient’s charts and investigations during index and return visits. If there was contention as to whether a diagnosis was considered wrong or delayed, a discussion was held among the three investigators to come to a consensus. Secondary outcomes include admission to the ED observation unit (EDOU) or ward on return visit. The data of this study was compared with findings of a previous study performed in the same institution, between 1 January 2005 and 30 June 2005 [11].

### Statistical Analysis

A formal sample size calculation was not performed, as this is a largely descriptive retrospective cohort study. Categorical variables are reported in proportions while continuous variables are reported in median with interquartile range (IQR) as appropriate. All data were populated in Microsoft Excel (Microsoft Corp, Redmond, WA, USA). Upon completion of data collection electronically, charts were reviewed for missing data, duplicate data and verification by two investigators independently (M.T.C., W.S.K.). The data were then exported to Stata 14 (StataCorp LP, College Station, TX, USA) for statistical analyses. Comparisons were made between URV patients with and without wrong or delayed diagnoses, and between patients who were admitted versus those who were not at their re-attendance visits. Differences in categorical variables were compared with a chi-square test or Fisher’s exact test. Skewed continuous variables were analyzed using the Mann–Whitney *U* test. Multivariate stepwise logistic regression was performed for variables with *p* < 0.10 derived from univariate analyses with odds ratios (OR) and 95% confidence intervals (CI) presented. Statistical significance was set at *p* < 0.05.

## 3. Results

### 3.1. Patient Characteristics

There were 67,422 unique patient attendances from 1 January 2014 to 30 June 2014 (Figure 1). Of these, 1298 (1.93%) were URVs from 1207 patients, indicating that some patients re-attended more than once within 72 h. Median age was 34 (IQR 24 to 52) years and 59.7% were male (Table 1).

There were 423 (35.0%) patients reviewed by senior ED doctors at index visits as compared to 589 (45.4%) during URVs (*p* < 0.001). Of the re-attendances, 469 (38.9%) had abnormal vital signs at initial triage, and 326 (27.0%) had a documented pain score of ≥5 (Table 2).

The most common presenting complaints were abdominal pain (22.2%), fever (21.0%), and gastrointestinal symptoms, including diarrhea, nausea, vomiting, and constipation (19.7%) (Table 3). There were 182 patients with other complaints, including 17 with palpitations, 22 with constitutional symptoms (loss of appetite, loss of weight, or lethargy), 14 with loss of consciousness or seizures, 10 with bleeding gastrointestinal tract, 14 with perianal complaints, and 6 with post-operative pain or bleeding. Out of 1298 URVs, a majority of 786 (60.6%) re-attended for persistent symptoms, of which 620 (47.8%) had persistent pain. Patients with abnormal vital signs (38.9%) and/or a pain score ≥5 (27%) at initial presentation were more likely to return (Table 2). Extension of medical leave (8.2%) and social reasons (1.2%) contributed to a minority of URVs.

### 3.2. Primary Outcome

One hundred ninety-one (15.8%) patients had an initial wrong or delayed diagnosis (κ = 0.85, 95% CI 0.81 to 0.90). There were six deaths; of which three were initially admitted to the ward or EDOU, one was discharged against medical advice, and two were discharged from the ED. Among them, five had delayed or wrong diagnoses during initial evaluation. The sixth patient discharged against medical advice during the index ED visit. One patient who died did not have any significant comorbidity and two had metastatic disease. Their ages ranged from 32 to 76 years.

Factors associated with initial wrong or delayed diagnosis are shown in Table 4. Lack of review by a senior ED physician at the index visit was not a significant predictor of wrong or delayed diagnosis (*p* = 0.73), regardless of whether the review was by a specialist (*p* = 0.70) or non-specialist (*p* = 0.64).

The variables were adjusted for factors with *p* < 0.10 after univariate analysis, including gender, age, initial disposition, initial primary doctor designation, alcohol use, psychiatric history, presentation with abdominal pain, trauma, fever, renal colic, neurological deficit, and nausea/vomiting. After multivariate stepwise logistic regression, the following were associated with initial wrong or delayed diagnosis (adjusted OR; 95% CI): presenting complaints of abdominal pain (2.99; 2.12 to 4.23), fever (1.60; 1.10 to 2.33), neurological deficit (4.26; 1.94 to 9.35), and initial disposition of discharge without follow-up (1.61; 1.15 to 2.26). URV patients between 21 and 30 years (0.60; 0.42 to 0.86), and those with presentations related to trauma (0.40; 0.18 to 0.90) were less likely to have an initial wrong or delayed diagnosis.

### 3.3. Secondary Outcome

Among URV patients, 459 (38.0%) required admission to either EDOU or inpatient wards. Factors associated with this are shown in Table 5. The variables were adjusted for factors with *p* < 0.10 after univariate analysis, including age, gender, initial primary doctor designation, review or discussion with a senior doctor, initial disposition, patient mobility, Charlson comorbidity index, presence of caregiver, significant comorbidities, presentation with abdominal pain, trauma, fever, upper respiratory symptoms, giddiness, vertigo, cellulitis/abscess, chest pain, gynecological complaints, ophthalmological complaints, asthma, nausea/vomiting, shortness of breath, musculoskeletal pain, initial vital signs, and patient acuity status. After logistic regression analysis, factors (adjusted OR; 95% CI) associated with admission were: male gender (1.88; 1.42 to 2.48); comorbidities of diabetes mellitus (2.07; 1.29 to 3.31), asthma (5.23; 1.59 to 17.26), and renal disease (7.48; 2.00 to 28.05); presenting complaints of abdominal pain (1.83; 1.32 to 2.55), fever (3.05; 2.10 to 4.44), and giddiness or vertigo (2.17; 1.26 to 3.73). Conversely, younger patients from 16 to 20 years (0.29; 0.13 to 0.62) and 21 to 30 years (0.49; 0.35 to 0.67) and patients that had a Charlson comorbidity index of 0 to 3 (0.25; 0.09 to 0.67); initial disposition of being discharged from the ED with (0.24; 0.15 to 0.38) or without (0.16; 0.10 to 0.26) follow-up; presenting complaints related to upper respiratory tract infection (0.45; 0.29 to 0.71), trauma (0.46; 0.27 to 0.77), or obstetrics and gynecology (0.39; 0.18 to 0.84) were less likely to be admitted.

### 3.4. Comparison with Results in 2005 Study

A 2005 study in the same center allowed us to reassess trends after 9 years [11]. URV rates have improved from 2005 (1.93%, versus 2.19% in 2005 (842/38,414), *p* = 0.006) despite a marked increase in patient attendances by 74% and reduction in admission rate from 36.2% in 2006 to 31.7% in 2014. Abdominal pain remains significant among URVs, with only a marginal decrease compared to 9 years ago (22.2% versus 25.1%). Similar to the previous study, younger patients aged 21 to 30 years constituted the majority of re-attendances but had a lower risk for wrong or delayed diagnoses (0.60; 0.42 to 0.86) and admission on return visits (0.49; 0.35 to 0.67). Patients who re-attended with fever continued to contribute to a large proportion (21.0%) of delayed diagnoses and were at a higher risk of admission (3.05; 2.10 to 4.44) on URVs.

## 4. Discussion

In our study, URVs comprised 1.93% of total patient attendances, similar to recent studies with the incidence rate of URVs ranging from 2.9% to 3.2% [12,13]. The majority of URVs were due to persistent symptoms and pain.

Patients with abnormal vital signs and/or a pain score ≥5 at initial presentation were more likely to return. This is similar to other studies showing that pain is a common initial presenting complaint that may be associated with re-attendance [7]. In our institution, patients with abnormal vitals or a pain score of at least moderate severity are triaged as PAC1 or 2 and are required to be reviewed by a senior ED physician. Although pain score has been incorporated into our ED triage since 2008, its follow-through for improvement or resolution of pain during the ED stay needs to be more closely monitored and documented. A standardized workflow has been developed for nurses to record vital signs and pain score just prior to discharge or transfer to the ward. The primary physician has to acknowledge the patient’s condition before completion of care in the ED.

The URV rate has improved since 2005, which could be due to the implementation of various strategies conceived from our previous study. These include more streamlined protocols for common complaints in the ED and EDOU, structured discharge advice sheets, and an early referral system for specialist care via rapid access clinics for services such as colorectal surgery, urology, hand surgery, and otolaryngology. The use of the EDOU for complaints such as chest pain has been shown to reduce re-attendance rates as well as reduce the cost to the health service [14]. In addition, increased senior ED physician staffing for better supervision of juniors may have indirectly impacted this improvement.

Abdominal pain remains significant among URVs, with only a marginal decrease compared to 9 years ago (22.2% versus 25.1%). Additional measures have been implemented since 2005, such as EDOU protocols for 24 h observation and serial evaluation of patients with abdominal pain and condition-specific structured discharge advice. Despite that, abdominal pain was independently associated with wrong or delayed diagnoses (2.99; 2.12 to 4.23) and admission (1.83; 1.32 to 2.55) during return visit. In recent years, ED point-of-care ultrasonography has been more commonly utilized for rapid identification of dangerous causes of abdominal pain (e.g., aneurysmal abdominal aorta, biliary tract abnormalities) [15]. ED physicians have been shown to be competent in bedside ultrasonography for specific conditions [16]. Since July 2014, our institution has developed a structured training program for bedside ultrasonography for residents and senior ED physicians. This will likely improve accuracy in diagnosing abdominal pain syndromes that can be evaluated by bedside ultrasonography.

Similar to the previous study, young patients aged 21 to 30 years constituted the majority of re-attendances but have a lower risk for wrong or delayed diagnoses (0.60; 0.42 to 0.86) and admission on return visits (0.49; 0.35 to 0.67). This may be due to increased health literacy among this age group, as compared to older counterparts [17]. More needs to be done in terms of education of this cohort of patients in management of symptoms and medical conditions.

An appreciable 15% of URVs demonstrated the primary outcome of either an initial wrong diagnosis or a delayed diagnosis. The presenting complaint of neurological deficit (4.26; 1.94 to 9.35) was most significantly associated with this. Of 31 patients who attended for neurological deficits, 9 (29.0%) were initially triaged to PAC3. Neurological conditions may have subtle and non-specific initial presentations which make diagnosis challenging [18]. New-onset neurological deficits should be evaluated more rapidly as treatment may be time-sensitive, such as in the case of stroke. Hence, these patients should be triaged to a higher acuity area in order to expedite consultation and perform thorough evaluation.

Patients who re-attended with fever continued to contribute to a large proportion (21.0%) of delayed diagnoses compared to 9 years ago and were at a higher risk of admission (3.05; 2.10 to 4.44) on their URVs. We postulate that patients may have initially presented early in their illness, where investigations were not clinically indicated or the source of fever was not obvious. Arranging follow-up care with primary physicians in the community may be a viable strategy to reduce URVs and wrong diagnosis [19]. Another strategy which has been implemented is giving patients discharge advice pamphlets on red flags in febrile illnesses, with the aim of decreasing adverse events as patients are better informed on when to seek medical advice.

Thirty-eight percent of patients required admission during the URVs. Factors independently associated were presentations with abdominal pain and fever. In addition, giddiness or vertigo (2.17; 1.26 to 3.73), and comorbidities such as diabetes mellitus and renal disease were also independently associated with admission on return visit.

An EDOU protocol for vertigo has been further streamlined in 2014 to incorporate vestibular rehabilitation and regular vestibular sedatives to decrease URVs. A possible strategy to manage patients presenting to the ED with vertigo with no indications necessitating admission would be the prescription of adequate vestibular sedatives, rapid access clinics with a vestibular rehabilitation specialist, and early follow-up in the outpatient setting [20]. Non-vertiginous giddiness remains a non-specific symptom, which can have underlying serious diagnoses. Therefore, it necessitates a more thorough evaluation and high threshold for discharge from the ED.

Patients with renal disease were most significantly associated with admission to the EDOU or ward on return visit (7.48; 2.00 to 28.05). Renal patients, especially those on dialysis are known to have increased risk of mortality and are increasingly being considered as immunocompromised [21]. They should be thoroughly evaluated in the ED prior to disposition as they may present atypically for various conditions, such as acute coronary syndrome.

The main strength of this study is that it involved a large sample from a tertiary hospital with all adult subspecialties, across all PAC statuses and shift timings. An extensive set of variables were collected in order to identify factors associated with URVs. The ability to evaluate effectiveness of strategies put in place from the previous study over a corresponding period in 2005 is also an advantage.

### Limitations

Our study has several limitations. First, being a single center study, our findings may not be representative of other EDs, which differ in patient demographics, sociocultural characteristics, physician practices, and healthcare resources. Second, the retrospective nature of the study is subjected to information bias from missing data in clinical documentation during index and return visits, especially information concerning socioeconomic status. We were also unable to evaluate communication of diagnosis, although the definition of diagnostic error used incorporates failure of communication to patients. Causal effects cannot be conclusively established. In order to minimize missing data, investigators reviewed all electronic records and hard copies of notes for data which could not be found electronically.

Next, we were unable to obtain data on specific investigations and interventions during the index visit, which may have affected the URV rate, as the broad scope of presenting complaints made it difficult to collect all available investigations. Nevertheless, we collected a wide range of variables, such as mobility, status of activities of daily living, and comorbidities, and believe that the trends found in our study are useful.

Third, while one of our aims was to compare with data from 2005, we were unable to gather the demographics of all patients who presented in 2005, as electronic data from that period was not accessible. Hence, we are unable to comment about secular trends and how the population has changed over this time period.

Fourth, we did not study re-attendances after 72 h, and re-attendances to other hospitals, which could have led to missed cases and unknown outcomes. However, previous studies have described URVs within 72 h of initial visit as a high risk for diagnostic and management errors [22]. Institutions such as the Centers for Disease Control and Prevention have also used 72 h as a benchmark in measuring URVs in their National Hospital Ambulatory Medical Care Survey [23].

Lastly, our definition of URV required similar presenting complaints for the first and subsequent visits. This may have resulted in inadvertent exclusion of URVs due to other presenting symptoms that developed from the same primary pathology. Consequently, we were only able to include in our results the patients’ presenting symptoms and complaints, rather than their eventual diagnoses at the initial visit and URV.

## 5. Conclusions

URVs with wrong or delayed diagnoses are associated with presenting complaints of neurological deficits, abdominal pain, fever, and discharge without follow-up. URVs resulting in admission are associated with abdominal pain, fever, giddiness or vertigo, and comorbidities of diabetes mellitus and renal disease. Strategies to reduce URVs include better education for younger patients, and early review of patients with abdominal pain syndromes by primary healthcare providers.

## Figures and Tables

**Figure 1 medicina-55-00457-f001:**
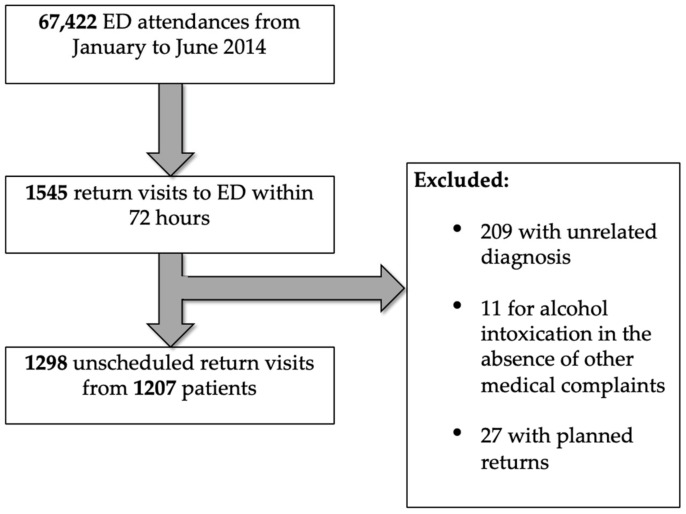
Selection of return visits to the emergency department (ED) within 72 h for analysis.

**Table 1 medicina-55-00457-t001:** Characteristics of patients who re-attended within 72 h.

Variable	Re-Attendance within 72 h (*N* = 1207)	All ED Attendances (*N* = 67,422)
Age, median (IQR)	34 (24–52)	41 (26–60)
Age group, years		
16 to 20	64 (5.3)	7066 (10.5)
21 to 30	420 (34.8)	16,246 (24.1)
31 to 40	231 (19.1)	10,636 (15.8)
41 to 50	164 (13.6)	8453 (12.5)
51 to 60	134 (11.1)	8882 (13.2)
61 to 70	106 (8.8)	6967 (10.3)
Above 70	88 (7.3)	9172 (13.6)
Male sex	721 (59.7)	40,489 (60.1)
Race		
Chinese	655 (54.3)	36,802 (54.6)
Malay	212 (17.6)	10,856 (16.1)
Indian	188 (15.6)	9517 (14.1)
Others	152 (12.6)	10,247 (15.2)
Activities of daily living status		
Independent	1197 (99.2)	
Assisted	10 (0.8)	
Mobility		
Independent	1188 (98.4)	
Ambulant with assistance	9 (0.7)	
Wheelchair	9 (0.7)	
Bedbound	1 (0.1)	
Presence of caregiver	292 (24.2)	
Comorbidities		
Diabetes mellitus (DM)	120 (9.9)	
DM with end organ damage	30 (2.5)	
Psychiatric history	73 (6.0)	
Ischemic heart disease	47 (3.9)	
Renal disease	39 (3.2)	
Cancer	33 (2.7)	
Cerebral vascular accident	31 (2.6)	
Congestive heart failure	17 (1.4)	
Peripheral vascular disease	12 (1.0)	
Liver disease	11 (0.9)	
Acquired immunodeficiency	8 (0.6)	
Chronic obstructive pulmonary disease	7 (0.6)	
Charlson comorbidity index		
0–3	1158 (95.9)	
4–5	25 (2.1)	
6–7	19 (1.6)	
8 and above	5 (0.4)	
Designation of primary doctor at index visit		
Resident/Medical officer	768 (63.6)	
Locum	390 (32.3)	
Senior resident	49 (4.1)	
Initial shift at presentation		
Morning (08:00 to 15:59)	487 (40.3)	31,650 (46.9)
Afternoon (16:00 to 21:59)	357 (29.6)	19,775 (29.3)
Night (22:00 to 07:59)	363 (30.1)	15,997 (23.7)
Initial disposition		
Discharged with follow-up	612 (50.7)	
Discharged without follow-up	442 (36.6)	
Discharged against advice	74 (6.1)	
Admission to EDOU	42 (3.5)	
Admission to ward	37 (3.1)	

Data are reported as *n* (%) unless otherwise stated. EDOU, emergency department observation unit; IQR, interquartile range.

**Table 2 medicina-55-00457-t002:** Abnormal vital signs at initial presentation of patients with re-attendances within 72 h.

Vital Sign	*n* (%)
Temperature ≥37.5 °C	145 (12.0)
Systolic blood pressure, mmHg	
above 140	214 (17.7)
below 90	8 (0.7)
Heart rate, beats per minute	
above 100	181 (15.0)
below 60	12 (1.0)
Respiratory rate above 20 breaths per minute	51 (4.2)
Oxygen saturation below 95%	6 (0.5)
Pain score 5 or higher	326 (27.0)

**Table 3 medicina-55-00457-t003:** Presenting complaints of patients with re-attendances within 72 h.

Presenting Complaint	*n* (%) *
Abdominal pain	288 (22.2)
Fever	273 (21.0)
Gastrointestinal	256 (19.7)
Upper respiratory tract infection	219 (16.9)
Musculoskeletal pain	166 (12.8)
Trauma	135 (10.4)
Headache	111 (8.6)
Lower back pain and sciatica	99 (7.6)
Chest pain	88 (6.8)
Giddiness and vertigo	83 (6.4)
Ophthalmological and Otolaryngological	69 (5.3)
Shortness of breath (excluding asthma)	67 (5.2)
Urological	63 (4.9)
Dermatological	62 (4.8)
Renal colic	48 (3.7)
Obstetric and Gynecological	48 (3.7)
Cellulitis and abscess	37 (2.9)
Neurological deficit	35 (2.7)
Psychiatric	32 (2.5)
Asthma	22 (1.7)
Others	182 (14.0)

*As patients may have presented with more than one presenting complaint, the total percentage may not add up to 100%.

**Table 4 medicina-55-00457-t004:** Factors significantly associated with wrong or delayed diagnosis after univariate analysis.

Factors	Wrong/Delayed Diagnosis, *n* (%)		Odds Ratios (OR) (95% CI)	*p* Value
	Yes (*N* = 191)	No (*N* = 1016)		
Age group, years				
21 to 30	53 (27.7)	367 (36.1)	0.68 (0.47 to 0.97)	0.026
51 to 60	30 (15.7)	104 (10.2)	1.63 (1.01 to 2.57)	0.027
Initial disposition				
Discharged with follow-up	80 (41.9)	532 (52.4)	0.66 (0.47 to 0.91)	0.008
Discharged without follow-up	88 (46.1)	354 (34.8)	1.60 (1.15 to 2.21)	0.003
Presenting complaint				
Abdominal pain	79 (41.4)	194 (19.1)	2.99 (2.12 to 4.20)	<0.001
Trauma	7 (3.7)	124 (12.2)	0.27 (0.11 to 0.59)	<0.001
Fever	54 (28.3)	207 (20.4)	1.54 (1.06 to 2.21)	0.015
Renal colic	1 (0.5)	41 (40.4)	0.13 (0.003 to 0.75)	0.015
Nausea/Vomiting	40 (20.9)	149 (14.7)	1.54 (1.02 to 2.30)	0.029
Neurological deficit	11 (5.8)	20 (2.0)	3.04 (1.29 to 6.79)	0.002

**Table 5 medicina-55-00457-t005:** Factors at the index visit significantly associated with admission to the EDOU or ward on re-attendance after univariate analysis.

Factor	Admission to EDOU or Ward, *n* (%)		OR (95% CI)	*p* Value
	Yes (*N* = 459)	No (*N* = 748)		
Age group, years				
16 to 20	9 (2.0)	55 (7.4)	0.25 (0.11 to 0.52)	<0.001
21 to 30	97 (21.1)	323 (43.2)	0.35 (0.27 to 0.46)	<0.001
41 to 50	79 (17.2)	85 (11.4)	1.62 (1.15 to 2.29)	0.004
51 to 60	68 (14.8)	66 (8.8)	1.80 (1.23 to 2.62)	0.001
61 to 70	60 (13.1)	46 (6.1)	2.29 (1.50 to 3.51)	<0.001
Above 70	64 (13.9)	24 (3.2)	4.89 (2.96 to 8.30)	<0.001
Male gender	229 (49.9)	257 (34.4)	1.90 (1.49 to 2.43)	<0.001
Presence of caregiver	133 (29.0)	159 (21.3)	1.51 (1.15 to 1.99)	0.002
Initial disposition				
Admission to ward	29 (6.3)	8 (1.1)	6.24 (2.75 to 15.91)	<0.001
Admission to EDOU	30 (6.5)	12 (1.6)	4.29 (2.10 to 9.29)	<0.001
Discharged without follow-up	120 (26.1)	322 (43.0)	0.47 (0.36 to 0.61)	<0.001
Discharged against advice	55 (12.0)	19 (2.5)	5.22 (3.00 to 9.44)	<0.001
Comorbidities				
Diabetes mellitus (DM)	79 (17.2)	41 (5.5)	3.58 (2.37 to 5.47)	<0.001
DM with end-organ damage	24 (5.2)	6 (0.8)	6.82 (2.69 to 20.54)	<0.001
Liver Disease	8 (1.7)	3 (0.4)	4.41 (1.05 to 25.88)	0.017
Congestive heart failure	14 (3.1)	3 (0.4)	7.81 (2.16 to 42.56)	<0.001
Ischemic heart disease	35 (7.6)	12 (1.6)	5.06 (2.53 to 10.82)	<0.001
Cerebral vascular accident	24 (5.2)	7 (9.4)	5.84 (2.41 to 16.16)	<0.001
Renal disease	36 (7.8)	3 (0.4)	21.13 (6.61 to 107.68)	<0.001
Cancer	22 (4.8)	11 (1.5)	3.37 (1.55 to 7.77)	<0.001
Acquired Immunodeficiency	6 (1.3)	1 (0.1)	9.89 (1.19 to 455.62)	0.009
Charlson comorbidity index				
0–3	416 (9.1)	742 (99.2)	0.08 (0.03 to 0.19)	<0.001
4–5	21 (4.6)	4 (0.5)	8.92 (2.98 to 35.90)	<0.001
6–7	17 (3.7)	2 (0.3)	14.35 (3.37 to 128.35)	<0.001
Presenting complaint				
Abdominal pain	130 (28.3)	143 (19.1)	1.67 (1.26 to 2.22)	<0.001
Trauma	27 (5.9)	104 (13.9)	0.39 (0.24 to 0.61)	<0.001
Fever	118 (25.7)	143 (19.1)	1.46 (1.10 to 1.95)	0.007
Upper respiratory tract infection	56 (12.2)	142 (19.0)	0.59 (0.42 to 0.84)	0.002
Giddiness/Vertigo	48 (10.5)	31 (4.1)	2.70 (1.65 to 4.46)	<0.001
Cellulitis and Abscesses	21 (4.6)	16 (2.1)	2.19 (1.08 to 4.54)	0.017
Chest Pain	41 (8.9)	41 (5.5)	1.69 (1.05 to 2.72)	0.021
Obstetric and gynecological	10 (2.2)	34 (4.5)	0.47 (0.20 to 0.98)	0.033
Ophthalmological	8 (1.7)	35 (4.7)	0.36 (0.14 to 0.80)	0.008
Asthma	14 (3.1)	6 (0.8)	3.89 (1.39 to 12.43)	0.003
Nausea or vomiting	96 (20.9)	93 (12.4)	1.86 (1.35 to 2.58)	<0.001
Shortness of breath (excluding asthma)	42 (9.2)	21 (2.8)	3.49 (1.98 to 6.28)	<0.001
Musculoskeletal pain	48 (10.5)	109 (14.6)	0.68 (0.47 to 0.99)	0.039
Otolaryngological	3 (0.7)	18 (2.4)	0.27(0.05 to 0.92)	0.024
Neurological deficits	20 (4.4)	11 (1.5)	3.05 (1.38 to 7.12)	0.002
Initial vitals sign abnormal	227 (49.5)	237 (31.7)	2.11 (1.65 to 2.70)	<0.001
Initial primary doctor				
Resident/Medical officer	255 (55.6)	513 (68.6)	0.57 (0.45 to 0.73)	<0.001
Locum	178 (38.8)	212 (28.3)	1.60 (1.24 to 2.06)	<0.001
Senior resident	26 (5.7)	23 (3.1)	1.89 (1.02 to 3.52)	0.027

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
