# Peer review of "Risk Factors for Emergency Department Unscheduled Return Visits"

_medicina, 2019, doi:10.3390/medicina55080457_

Round 1

Reviewer 1 Report

The aim of study is to identify reasons for unscheduled return  visits (URVs), and risk factors for diagnostic errors leading to URVs, with comparisons to data from 9 years ago. It should be noted that the study conducted 9 years ago was described in the publication (Ref. 15), however quality of the statistical analysis of the data in this article is very low, some data is missing (see the limitation section) so comparing the data may be inaccurate. 

The design of the study is interesting, however some methodological issues are unclear. The study is of sufficient size and duration to reveal data about URVs in the particular hospital studied.

Revisits are evaluated without analyzing the circumstances of arrival; however there are different reasons for revisits. In addition to the patient deciding to revisit the ED sometimes ED doctors will ask patients to come back for a recheck; the primary care physician can also send a patient back to the ED for reassessment,  a certain test or admission; sometimes patients come back because of the lack of accessibility of primary care physician. The authors must address this issue.  

Some definitions should be explained, such as  "index visit", "Patient acuity category".

Other specific definitions are required:  how were the presenting complaints developed? It is unclear how they were derived for example, would a bronchitis fall under upper respiratory tract infection or signs and symptoms (fever, shortness of breath); would a suspected meningitis fall under neurological deficits, or other (fever, headache); where the diarrhea assigned?

In the Results section would be helpful to see additional data with comparison the initial and URV diagnosis, not only symptoms.

In the Discussion section the observations are not supported by the data or literature sufficiently.

Significant limitations mentioned in the manuscript possibly undermine the results of the study.

Some references are very old. Even 9 of them are older than 10 years. I would recommend leaving only the most important ones.

Reviewer 2 Report

This paper has broad appeal, not just to hospital ED clinicians but also those working in primary care and EMS environments.

The rational for not undertaking a sample size calculation is reasonable; however it may be worth explaining the intent of the statistical analysis short of generalisation.

Was the 2005 study conducted over the same time period ie Jan through Jun? The comparison between 2014 and 2005 appears to be rather important, but this only amounts to a paragraph in the results section. I would have preferred to see a more detailed analysis. There appears to be a mixed message regarding the analysis of the 2005 study and comparison with the 2014 findings. This seems to have greater significance in the title but less emphasis in the body of the work. It almost feels like an add on in the body of the work and not an integral part of the study. I would suggest you rationalise this apparent conflict and make clear to the reader what your intentions of this study are.

Specific comments

The abstract makes no reference to the comparison with the 2005 study, yet this is a key part of the body of work and title.`

Round 2

Reviewer 1 Report

Thank you for taking into account the comments, the manuscript is now clearer, but patients grouped on the basis of complaints is not very clear.

Author Response

Reviewer 1:

Thank you for taking into account the comments, the manuscript is now clearer, but patients grouped on the basis of complaints is not very clear.

Response 1: 

Dear reviewer, thank you very much for the comments. In order to elaborate further on the grouping of complaints and to make it clearer for the readers, we have created a Supplementary Table to explain how the various complaints are grouped. We have referenced it in Line 87 of the manuscript.